# Increased Adipose Tissue Expression of IL-23 Associates with Inflammatory Markers in People with High LDL Cholesterol

**DOI:** 10.3390/cells11193072

**Published:** 2022-09-29

**Authors:** Shihab Kochumon, Amal Hasan, Fatema Al-Rashed, Sardar Sindhu, Reeby Thomas, Texy Jacob, Amnah Al-Sayyar, Hossein Arefanian, Ashraf Al Madhoun, Ebaa Al-Ozairi, Fawaz Alzaid, Heikki A. Koistinen, Fahd Al-Mulla, Jaakko Tuomilehto, Rasheed Ahmad

**Affiliations:** 1Department of Immunology & Microbiology, Dasman Diabetes Institute, Dasman 15462, Kuwait; 2Animal and Imaging Core Facilities, Dasman Diabetes Institute, Dasman 15462, Kuwait; 3Medical Division, Dasman Diabetes Institute, Dasman 15462, Kuwait; 4Institut Necker Enfants Malades (INEM), French Institute of Health and Medical Research (INSERM), Immunity & Metabolism of Diabetes (IMMEDIAB), Université de Paris Cité, 75014 Paris, France; 5Department of Medicine, Helsinki University Hospital, University of Helsinki, 00029 Helsinki, Finland; 6Department of Public Health and Welfare, Finnish Institute for Health and Welfare, 00271 Helsinki, Finland; 7Minerva Foundation Institute for Medical Research, 00290 Helsinki, Finland; 8Department of Genetics and Bioinformatics, Dasman Diabetes Institute, Dasman 15462, Kuwait; 9Department of Public Health, University of Helsinki, 00100 Helsinki, Finland

**Keywords:** IL-23, LDL-cholesterol, adipose tissue, cytokines/chemokines, inflammation

## Abstract

Chronic low-grade inflammation induced by obesity is a central risk factor for the development of metabolic syndrome. High low-density lipoprotein cholesterol (LDL-c) induces inflammation, which is a common denominator in metabolic syndrome. IL-23 plays a significant role in the pathogenesis of meta-inflammatory diseases; however, its relationship with LDL-c remains elusive. In this cross-sectional study, we determined whether the adipose tissue IL-23 expression was associated with other inflammatory mediators in people with increased plasma LDL-c concentrations. Subcutaneous adipose tissue biopsies were collected from 60 people, sub-divided into two groups based on their plasma LDL-c concentrations (<2.9 and ≥2.9 mmol/L). Adipose expression of IL-23 and inflammatory markers were determined using real-time qRT-PCR; plasma concentrations of total cholesterol (TC), triglyceride (TG), high-density lipoprotein cholesterol (HDL-c) and LDL-c were determined using the standard method; and adiponectin levels were measured by enzyme-linked immunosorbent assay (ELISA). Adipose IL-23 transcripts were found to be increased in people with high LDL-c, compared to low LDL-c group (H-LDL-c: 1.63 ± 0.10–Fold; L-LDL-c: 1.27 ± 0.09–Fold; *p* < 0.01); IL-23 correlated positively with LDL-c (r = 0.471, *p* < 0.0001). Immunochemistry analysis showed that AT IL-23 protein expression was also elevated in the people with H-LDL-c. IL-23 expression in the high LDL-c group was associated with multiple adipose inflammatory biomarkers (*p* ≤ 0.05), including macrophage markers (CD11c, CD68, CD86, CD127), TLRs (TLR8, TLR10), IRF3, pro-inflammatory cytokines (TNF-α, IL-12, IL-18), and chemokines (CXCL8, CCL3, CCL5, CCL15, CCL20). Notably, in this cohort, IL-23 expression correlated inversely with plasma adiponectin. In conclusion, adipose IL-23 may be an inflammatory biomarker for disease progression in people with high LDL-c.

## 1. Introduction

Interleukin (IL)-23 is a heterodimeric pro-inflammatory cytokine, produced by various immune cells. IL-23 is mainly secreted by activated monocytes, macrophages and dendritic cells found in peripheral tissues such as skin, intestinal mucosa and lungs [1]. It can also be secreted by other cells, including innate lymphoid cells, γδ T lymphocytes and B cells [2,3]. IL-23 is reported to be involved in the pathogenesis of several autoimmune inflammatory disorders, including colitis, gastritis, psoriasis, and arthritis [4,5,6,7]. In rheumatoid arthritis patients, IL-23 levels have been found to be elevated in the plasma, synovial fluid and synovial tissue of the affected joints [8,9]. IL-23 is known to have pro-osteoclastogenesis effects by promoting inflammation and bone destruction through the interaction with other cytokines, such as IL-17 and tumor necrosis factor (TNF)-α[10]. Substantial evidence supports that these cytokines also regulate mucosal inflammatory responses in the gut and play key roles in the pathophysiology of several diseases, including inflammatory bowel disease, ulcerative colitis and Crohn’s disease [11,12,13,14]. Notably, IL-23 also plays a leading role in innate and adaptive immunity by activating NK cells, enhancing T-cell proliferation and regulating antibody production [15,16].

Increased low-density lipoprotein cholesterol (LDL-c) levels contribute causally to the induction of inflammation and atherosclerosis. Lowering of LDL-c concentration is, therefore, the main goal to prevent progression of atherogenesis and to alleviate associated inflammation [17,18,19]. The lipid-rich plaques harbor inflammatory immune cell populations, including neutrophils and mononuclear phagocytes [20]. Anti-inflammatory agents, such as colchicine, are known to have therapeutic effects in patients with familial Mediterranean fever and other rheumatic and non-rheumatic diseases, especially by inhibiting neutrophil functions [21]. LDL-c induces activation of inflammasome, leading to the production of IL-1β and IL-18, which in turn accelerate atherosclerosis. Multiple diseases have been linked to enhanced IL-1β and IL-18 and/or inflammasome activation [22]. An association between IL-23 and disease progression in patients with carotid atherosclerosis has been demonstrated; reducing levels of IL-23 and LDL-c suppressed inflammation and improved atherosclerosis [23].

Plasma IL-23 levels were found to positively correlate with plasma LDL-c and triglycerides, supporting the notion of “bioumoral bridges” regulating the complex crosstalk between atopy and hyperlipidemia [24]. Nonetheless, the relationship between LDL-c and adipose tissue expression of IL-23 has not yet been investigated. We hypothesized that adipose IL-23 expression could be positively modulated in people with increased plasma LDL-c levels, and these changes might be consistent with an inflammatory profile in the adipose tissue. Here, we report that people with increased plasma levels of LDL-c display high expressions of IL-23 in the adipose tissue, which correlates positively with adipose expression of inflammatory macrophage markers, TLR2, IRF3, and pro-inflammatory cytokines/chemokines, suggesting IL-23 as a potential biomarker of metabolic inflammation in people with high circulatory LDL-c.

## 2. Materials and Methods

### 2.1. Study Population and Anthropometric Measurements

A cohort of 60 people without diabetes was enrolled in this study. Written informed consent was obtained from all study participants, and the study was approved (RA# 2010–003; June 2010) by the Dasman Diabetes Institute ethics committee, which follows the updated guidelines and ethical principles for medical research involving human subjects as per the WMA declaration of Helsinki. Anthropometric measurements were done using standard methods. Whole-body composition (percent body fat, lean mass, and total body water) was measured by using a IOI353 Body Composition Analyser (Jawon Medical, Seoul, Korea). BMI was calculated using a standard formula: BMI = body weight (kg)/height (m^2^). The cohort was divided into two groups based on their plasma LDL-c concentrations: the low LDL-c group (LDL-c: <2.9 mmol/L) and high LDL-c group (LDL-c: ≥2.9 mmol/L); the cutoff of 2.9 mmol/L refers to the mid-point of the range (2.6–3.3 mmol/L) of near-optimal LDL-c levels that corresponds to a risk for developing symptomatic cardiovascular disease events [25]. The characteristics of the participants are summarized in Table 1.

### 2.2. Collection of Subcutaneous Adipose Tissue

Adipose tissue samples, weighing ~0.5 g, were collected from people without diabetes following abdominal (subcutaneous) fat biopsy, lateral to the umbilicus, using a standard aseptic surgical procedure. Briefly, the area was sterilized using an alcohol swab, and local anesthesia was induced using 2% lidocaine (2 mL) injection. The subcutaneous fat pad was collected through a small skin incision, about 0.5 cm in size [26]. After collection, the sample was incised into smaller pieces, rinsed in cold PBS, fixed in 4% paraformaldehyde for 24 h, and then embedded in paraffin. The adipose tissue samples were also placed in RNAlater and stored at −80 °C until further use [27].

### 2.3. Measurement of Biochemical Parameters

Peripheral blood samples were obtained from following overnight fasting, and the samples were analyzed for fasting plasma glucose (FPG), lipid profile, glycated hemoglobin (HbA1c), fasting insulin and adiponectin. Glucose and lipid profiles (triglycerides, LDL-c, HDL-c and total cholesterol levels) were measured by using Siemens Dimension RXL chemistry analyzer (Diamond Diagnostics, Holliston, MA, USA). The Friedewald method was employed for assessing LDL-c concentrations [28]. HbA1c was determined using the Variant device (BioRad, Hercules, CA, USA). Plasma adiponectin was assessed using immunobead assays (Luminex, Austin, TX, USA). All assays were carried out following the instructions of the manufacturers.

### 2.4. Real-Time qRT-PCR

Total RNA was isolated from the adipose tissue (80 mg) using an RNeasy kit (Qiagen, Valencia, CA, USA) and following the instructions of the manufacturer. RNA was reverse transcribed into a cDNA template as instructed (HighCapacity cDNA Reverse Transcription kit; Applied Biosystems, Foster City, CA, USA). For RT-PCR, cDNA (50 ng of each sample) was amplified (40 cycles) using TaqMan Gene Expression Master Mix (Applied Biosystems, Foster City, CA, USA) and gene-specific TaqMan gene expression assays (Applied Biosystems, Foster City, CA, USA) (Table 2) on a 7500 Fast Real-Time PCR System (Applied Biosystems, CA, USA). Each amplification cycle involved denaturation (15 s at 95 °C) and annealing/extension (1 min at 60 °C) after uracil DNA glycosylases (UDG) activation (2 min at 50 °C) and AmpliTaq gold enzyme activation (10 min at 95 °C). GAPDH expression was used as an internal control for normalization of the data obtained. Expression of each target gene relative to the control was calculated using the standard 2^−ΔΔCt^ method [29,30,31].

### 2.5. Immunohistochemistry (IHC)

IHC was performed to determine protein expression in the adipose tissue. Paraffin-embedded, 4 µm-thick adipose tissue sections were processed as described elsewhere [26]. Briefly, adipose tissue sections were incubated with primary antibodies, i.e., 1:200 dilution of rabbit polyclonal anti-IL-23 antibody (Abcam^®^ ab115759; Waltham, MA, USA), 1:200 dilution of rabbit polyclonal anti-TNF antibody (Novus Biologicals Centennial, CO, USA; NBP1–19532) and 1:500 dilution of rabbit polyclonal anti-CCL5 antibody (R&D Systems AF478) overnight at room temperature. After washing thrice with PBS (0.5% Tween), slides were treated with secondary antibody (goat anti-rabbit conjugated with horseradish peroxidase (HRP) polymer chain; EnVision™ Kit from Dako, Glostrup, Denmark) for 1 h, and color was developed using 3,3ʹ-diaminobenzidine (DAB) chromogen substrate. Samples were washed in running tap water, lightly counterstained with Harris hematoxylin, dehydrated through ascending grades of ethanol (75%, 95% and 100 %), cleared in xylene and finally mounted in dibutylphthalate xylene (DPX). For analysis to assess the regional heterogeneity in tissue samples, digital photomicrographs of four different regions were selected at 20× magnification using PannoramicScan (3DHistech, Budapest, Hungary). All samples were analyzed using image J software (NIH, Bethesda, MA, USA) and % area of IHC staining was calculated using Image J software.

### 2.6. Statistical Analysis

GraphPad Prism software (La Jolla, CA, USA) and SPSS for Windows version 19.01 (IBM SPSS Inc., Chicago, IL, USA) were used for performing statistical analysis. Data are shown as mean ± standard deviation values, unless otherwise indicated. Two-tailed unpaired Student’s *t*-test was used for group means comparison. Spearman correlation analysis was performed to determine associations between different variables. For all analyses, *p*-values < 0.05 were considered significant.

## 3. Results

### 3.1. High IL-23 Expression Levels in Adipose Tissue of People with Increased Plasma Levels of LDL-C

*IL-23* gene expression is modulated in various inflammatory disorders. We asked whether plasma LDL-c levels correlated with the adipose IL-23 expression. To this end, we found that *IL-23* gene expression in the adipose tissue was significantly elevated (*p* = 0.011) in people with increased LDL-c levels (1.63 ± 0.35-fold) compared to people with lower LDL-c concentrations (1.28 ± 0.14-fold) (Figure 1A).

IL-23 gene expression correlated positively with LDL-c (r = 0.471, *p* = 0.0001; Figure 1B). IHC analysis showed that adipose IL-23 protein levels were also elevated (*p* < 0.0001) in the people with high LDL-c levels (Figure 2A,B). IHC isotype negative control and positive control for IL-23 is shown in Appendix A. 

### 3.2. Increased IL-23 Gene Expression in the Adipose Tissue Relates to Macrophage Markers

Adipose tissue macrophages, especially the classically activated M1 macrophages, are the major contributors to metabolic inflammation in this compartment [32]. Next, we aimed to determine whether the elevated adipose IL-23 mRNA expression in people with high LDL-c was associated with the macrophage markers expression in this compartment. To this end, we found that adipose IL-23 mRNA expression in the H-LDL-c group associated positively with the gene expression of several macrophage markers, including CD11c (r = 0.56; *p* = 0.0004), CD16 (r = 0.59; *p* = 0.0001), CD68 (r = 0.39; *p*= 0.024), CD86 (r = 0.37; *p* = 0.02) and CD127 (r = 0.57; *p* = 0.0004) (Table 3). However, regarding people with L-LDL-c, adipose *IL-23* mRNA expression was found to associate only with CD16 (r = 0.48; *p* = 0.03) and CD163 (r = 0.53, *p* = 0.01).

### 3.3. Increased Adipose IL-23 Gene Expression in People with High LDL-C Relates to TLR2-IRF3 Axis

Increased TLR expression or activity in adipose tissue is involved in metabolic inflammation [33,34]. Next, we aimed to determine whether the elevated *IL-23* mRNA expression in adipose tissue from people with high LDL-c was concordant with TLRs and/or TLR-downstream signaling partner(s). As shown in Table 4, adipose *IL-23* expression in people with high LDL-c levels associated positively with the gene expression of TLR2 (r = 0.49; *p* = 0.008). However, we did not find association of *IL-23* mRNA expression and any other TLR. Furthermore, *IL-23* gene expression was positively correlated with expression of an inflammatory transcription factor IRF3 (r = 0.46, *p* < 0.01), while there was no association found with anti-inflammatory transcription factor IRF4.

### 3.4. Increased IL-23 Gene Expression in the Adipose Tissue Relates to Inflammatory Cytokines and Chemokines

The relationship between adipose tissue *IL-23* gene expression and other markers of adipose inflammation in people with H-LDL-c levels remains unclear. Next, we asked whether the elevated adipose *IL-23* mRNA expression in H-LDL-c people was concordant with expression of other inflammatory immune markers, including proinflammatory cytokines/chemokines in this compartment. In this regard, as shown in Table 5, *IL-23* gene expression associated positively with several proinflammatory cytokines and chemokines, including IL-12A (r = 0.52; *p* = 0.004), IL-18 (r = 0.35; *p* = 0.043), TNF-α (r = 0.38; *p* = 0.032), CCL3 (r = 0.43; *p* = 0.017), CCL5 (r = 0.63; *p* = 0.0001), CCL15 (r = 0.37; *p* = 0.02), and CCL20 (r = 0.34; *p* = 0.049); and also including TGF-β (r = 0.40, *p* = 0.018) and IFN-β1 (r = 0.40, *p* = 0.024). However, no such associations were identified between adipose *IL-23* mRNA expression and these cytokines/chemokines in L-LDL-c people, in which *IL-23* gene expression was found to associate only with a TH1 cytokine IL-2 (r = 0.504; *p* = 0.017). Furthermore, to confirm the expression of protein levels in the adipose tissue, we performed immunohistochemistry analysis on two key inflammatory markers as representatives. Immunohistochemistry analysis showed that TNF-α (*p* = 0.0042; Figure 3A,B) and CCL5 (*p* = 0.0053; Figure 4A,B) were significantly upregulated in the people with H-LDL-c compared to people with L-LDL-c. Our protein data show that TNF-α positively correlated with IL-23 (r = 0.721; *p* = 0.0187; Figure 3C). Similarly, CCL5 protein was positively correlated with IL-23 (r = 0.823; *p* = 0.004; Figure 4C).

### 3.5. Adipose IL-23 Gene Expression in People with H-LDL-C Associates Inversely with Plasma Adiponectin Levels

Adiponectin plays an important role in the metabolism and regulation of lipid profile, and it is secreted by the adipose tissue. It is known to be downregulated in states of obesity and insulin resistance [35]. Next, we asked whether the changes in adipose tissue *IL-23* gene expression in H-LDL-c people associated with the metabolic markers, such as plasma levels of triglycerides, total cholesterol, HDL-c, LDL-c, fasting plasma glucose, HbA1c, insulin and adiponectin. As shown in Table 6, *IL-23* gene expression was found to be inversely associated with adiponectin levels (*r* = −0.44, *p* < 0.037) in the H-LDL-c group, while *IL-23* gene expression had no association with any of these metabolic markers in the L-LDL-c group. 

Volcano plots represent the correlation of IL-23 with differentially expressed genes in adipose tissue of people with L-LDL-c and H-LDL-c (Figure 5A,B).

## 4. Discussion

IL-23 is a novel cytokine, identified as a member of the IL-12 family. IL-23 has been associated with a number of autoimmune inflammatory conditions, such as colitis, gastritis, psoriasis and arthritis [4,5,6,7]. IL-23 is present in the plasma, synovial fluid and synovial tissue of patients with rheumatoid arthritis, whereas it is absent in healthy joints [8,9]. IL-23 regulates a variety of pathobiological effects that are associated with the advancement of different cardiovascular diseases including hypertension, atherosclerosis, cardiac hypertrophy, myocardial infarction, aortic dissection, and acute cardiac injury [36]. Elevated LDL-c has been identified as a major cause of coronary heart disease [37].

The changes in the adipose tissue expression of IL-23 and their relationship with various immunometabolic biomarkers in this compartment in the metabolic disease setting remain elusive. Enhanced *IL-23* mRNA expression has been identified in the peripheral blood mononuclear cells in type-2 diabetic patients [38]. Nonetheless, it remains unclear how the adipose tissue IL-23 expression is modulated in relation to the plasma lipid profile, especially the LDL-c concentrations. Herein, we show, for the first time to our knowledge, that *IL-23* gene expression was significantly upregulated in the adipose tissues of the people with increased plasma LDL-c concentrations as compared to those with normal or low LDL-c concentrations. Furthermore, *IL-23* gene expression associated positively with plasma concentrations of LDL-c. Inflammation and LDL-c dysregulation are two closely interrelated contributors to the atherosclerosis pathogenesis. Lowering LDL-c concentrations has possible pleiotropic anti-inflammatory effects that also improve atherosclerosis [22]. Increased plasma IL-23 levels were reported in patients with hypertension [39]. However, no clear regulatory role or association of LDC-c with the adipose tissue expression of IL-23 has yet been described. We herein report that LDL-c correlates positively with the adipose tissue expression of IL-23. The role of inflammation is well known in the progression of atherosclerosis. In atherosclerotic plaques, infiltration of several types of immune effector cells, including T cells and macrophages, has been reported [40]. In cardiovascular disease, macrophages play a central role in the development of plaques. Relating to macrophage’s role in tissue inflammation, classically activated M1 macrophages are implicated in the initiation and sustenance of inflammatory responses, while the alternatively activated M2 macrophages are linked to inflammation resolution [41]. Although LDL-c has been implicated in atherosclerotic plaque formation [41,42], clarifying the plausible influence of LDL-c on the adipose tissue macrophages remains a major concern. Our data show that LDL-c levels are positively correlated with proinflammatory macrophages in the adipose tissue expressing CD11c and CD16 surface markers. Of note, inflammatory macrophages are also known to express certain TLRs and associated downstream signaling molecules, enabling these immune effector cells to attach to cognate pathogen-associated molecular patterns or respond to the presence of danger-associated molecular patterns. In this regard, LDL-c was found to affect expression of the pattern-recognition receptor TLR2 and lead to the phosphorylation of key signaling kinases involved in cell activation [43]. Interestingly, we found in our study that *IL-23* expression was correlated with *TLR2* and *IRF3* expression in the adipose tissue of the people with increased plasma levels of LDL-c. Consistent with our observation, at least in part, of increased *TLR2* expression linked with high LDL-c levels, a pro-atherogenic impact of TLR2 activation has been reported via induction of intimal hyperplasia and development of atherosclerotic lesions [44]. In a mouse model study, Mullick et al. showed that TLR2 was important in atherosclerosis, and the exogenous TLR2 exposure enhanced atherosclerosis [45]. Previous studies also highlighted the role of TLR-downstream signaling transcription factor IRF3 in metabolic inflammation [46,47,48,49], which is concordant with our data of increased adipose *IRF3* expression in people with elevated LDL-c; however, these studies have not explored the association between IL-23 and IRF3 in adipose tissues of people with high LDL-c levels. Notably, as expected, no association was found between IL-23 and IRF4, considering consensus on its role as an anti-inflammatory transcription factor [50].

The white adipose tissue is a site for excessive energy storage, and it is also an active endocrine organ that secretes a wide variety of adipocytokines [51]. Having found a correlation between adipose tissue IL-23 and increased inflammatory macrophage marker expression in people with H-LDL-c, we next determined how the changes in adipose *IL-23* expression related to the local expression of inflammatory cytokines and chemokines. To this end, we found that *IL-12*, *IL-18* and *TNF-α* transcript expression was significantly enhanced, which correlated positively with the adipose *IL-23* expression in people with high LDL-c levels.

These data suggest that increased LDL-c levels may be the predictor of increased adipose tissue expression of IL-23. The proinflammatory cytokines IL-12, IL-18, and TNF-α are expressed by various immune cells, including monocytes, macrophages, dendritic cells, neutrophils, and endothelial and epithelial cells in the human adipose tissue, and these inflammatory cytokines have been implicated in the pathogenesis of a number of inflammatory disorders, including atherosclerosis, coronary heart disease, and type-2 diabetes, in all of which these cytokines have been found to be consistently elevated [52,53].

The chemokine system plays an important role in the pathophysiology of cardiometabolic diseases. Chemokines are small-sized chemotactic signaling proteins released by activated leukocytes and may lead to the induction of various proinflammatory cytokines, including IL-1β, TNF-α and IL-6 [54,55]. Changes in chemokine expression have been linked with LDL-c, development of atherosclerosis, and exacerbation of cardiovascular disease [56]. Our results indicate that the adipose tissue expression of several CC-chemokines, including *CCL3, CCL5, CCL15* and *CCL20,* was positively correlated with adipose *IL-23* expression in people with high HDL-c. Notably, chemokines CCL3, CCL5 and CCL18 were identified as the risk factors of short-term mortality in patients with acute coronary syndromes [57,58]. Regarding these chemokines, CCL3 deficiency inhibits atherosclerotic lesion development by affecting neutrophil accumulation [59]. CCL5 expression is modulated by LDL-c levels [60]. Atherogenic LDL induces the CCL18 expression in primary human monocyte-derived macrophages [61]. Consistent with our data, previous studies show that CCL3 and CCL5 are associated with LDL-c and atherosclerosis. Our data extrapolate these findings by showing that IL-23 is associated positively with these inflammatory biomarkers in the adipose tissue in people with high LDL-c concentrations.

## 5. Conclusions

In conclusion, our data show that adipose tissue *IL-23* gene expression is increased in people with high LDL-c levels, and this immune modulation is consistent with a wide spectrum of adipose inflammatory signatures, suggesting that IL-23 may be an important biomarker for predicting metabolic inflammation and cardiovascular disease progression.

## Figures and Tables

**Figure 1 cells-11-03072-f001:**
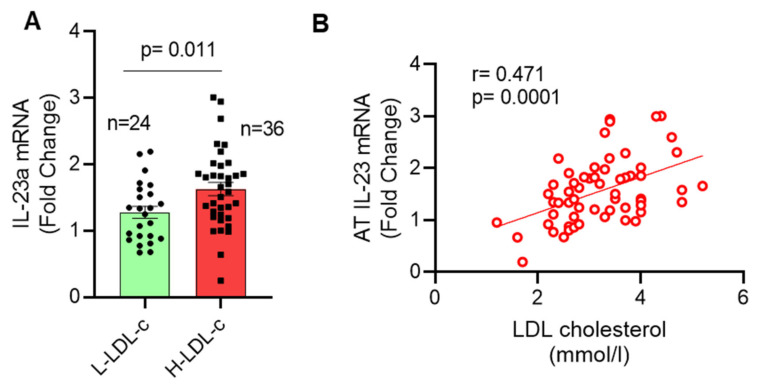
Increased adipose tissue *IL-23* gene expression in people with high plasma LDL-c. Adipose tissue samples were obtained from 60 people and divided into two groups based on their plasma LDL-c levels as high LDL-c (<2.9 mmol/L) and low LDL-c group (≥2.9 mmol/L). Total cellular RNA was isolated from adipose tissue, and *IL-23* gene expression was determined by real time qRT-PCR. Relative mRNA expression compared to GAPDH expression was presented as fold change. (**A**) IL-23 levels in each group are shown (bar graph). Each dot represents the individual value of IL-23, and the line represents the mean value. (**B**) Spearman’s correlation between *IL-23* gene expression and plasma LDL-c (mmol/L) is shown. Data are represented as mean ± SEM values. Group means were compared using two-tailed unpaired Student’s *t*-test. A *p*-value of < 0.05 was considered as statistically significant.

**Figure 2 cells-11-03072-f002:**
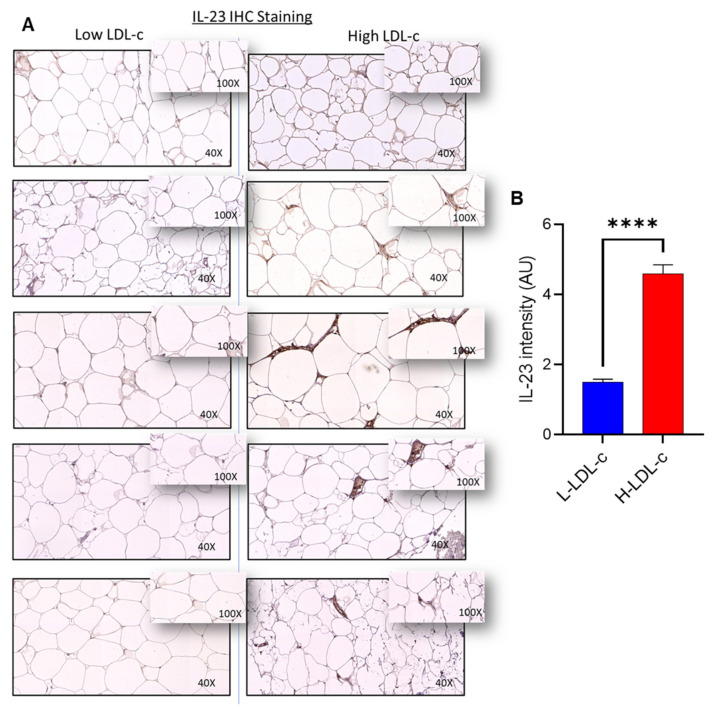
Increased adipose tissue IL-23 protein expression in people with high LDL-c. IL-23 protein expression was assessed in people with low LDL-c (L-LDL-c) and high LDL-c (H-LDL-c) levels, 5 each, using IHC, as described in “Materials and methods”. (**A**) The representative IHC images obtained from three independent determinations show the elevated adipose tissue expression of IL-23 protein (40× or insert 100× magnification) in people with H-LDL-c (≥2.9 mmol/L) compared to those with L-LDL-c (˂2.9 mmol/L). (**B**) Increased IL-23 protein expression is shown in H-LDL-c group compared to L-LDL-c group (*p* < 0.0001). Staining intensity is expressed as arbitrary units (AU), determined based on Aperio-positive pixel counts (Aperio software algorithm version 9.0). IL-23 protein expression data are presented as mean ± SEM values. All *p*-values < 0.05 were considered statistically significant. **** Represent highly significant.

**Figure 3 cells-11-03072-f003:**
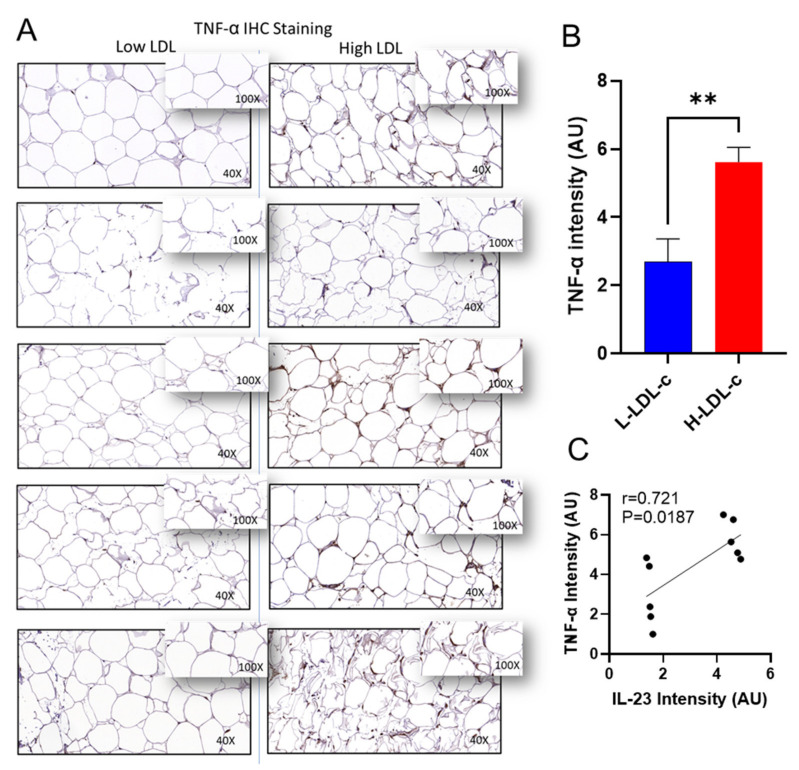
Increased adipose TNF-α protein expression in people with high LDL-c and positively correlated with IL-23 protein. TNF-α protein expression was determined in people with low LDL-c (L-LDL-c) and high LDL-c (H-LDL-c), five each, using IHC. (**A**) The representative IHC images obtained from three independent measurements with similar results show the elevated adipose tissue TNF-α expression (40× or insert ×100 magnification) in people with H-LDL-c (≥2.9 mmol/L) compared to those with L-LDL-c (˂2.9 mmol/L). (**B**) Elevated TNF-α protein expression is presented in the H-LDL-c group compared to the L-LDL-c group (*p* = 0.0042). Staining intensity shown as arbitrary units (AU) was determined based on Aperio-positive pixel counts (Aperio software algorithm, version 9.0). (**C**) IL-23 protein correlated with TNF-α protein in adipose tissue. *p*-values < 0.05 were considered as statistically significant. ** Highly significant.

**Figure 4 cells-11-03072-f004:**
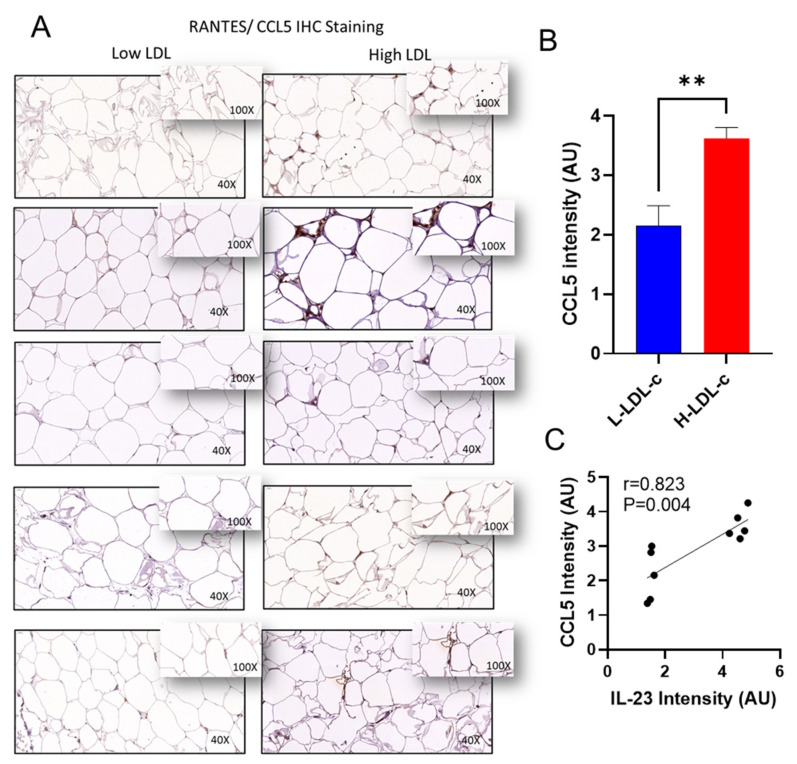
Increased adipose CCL5 protein expression in people with high LDL-c and positively correlated with IL-23 protein. CCL5 protein expression was determined in people with low LDL-c (L-LDL-c) and high LDL-c (H-LDL-c), five each, using IHC. (**A**) The representative IHC images obtained from three independent determinations with similar results show the increased adipose tissue TNF-α expression (40× or inset ×100 magnification) in people with H-LDL-c (≥2.9 mmol/L) compared to those with L-LDL-c (˂2.9 mmol/L). (**B**) Increased CCL5 protein expression is shown in the H-LDL-c group compared to the L-LDL-c group (*p* = 0.0053). Staining intensity shown as arbitrary units (AU). (**C**) IL-23 protein correlated with CCL5 protein in adipose tissue. *p*-values < 0.05 were considered as statistically significant. ** Highly significant.

**Figure 5 cells-11-03072-f005:**
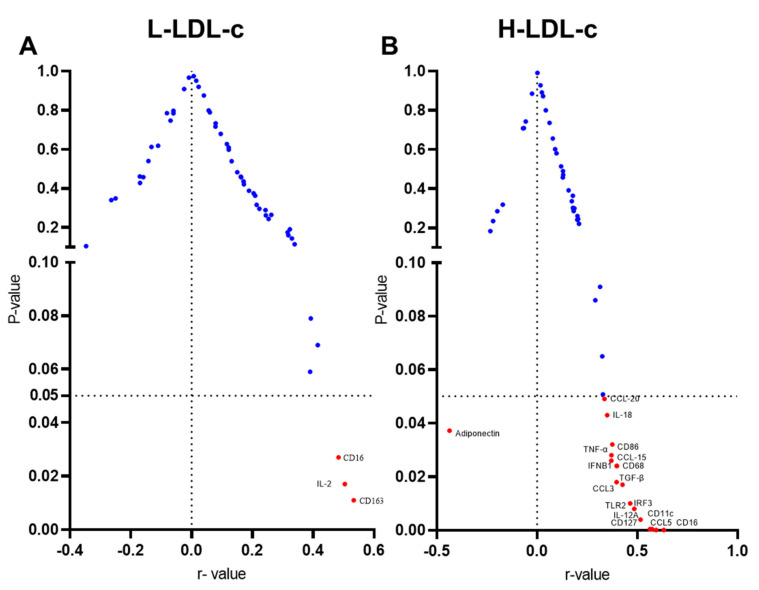
(**A**) Volcano plots represent correlation of IL-23 with differentially expressed genes in adipose tissue of people with L-LDL-c; red dots represent the significant correlation of IL-23 with other markers. (**B**) Volcano plots represent correlation of IL-23 with differentially expressed genes in adipose tissue of people with H-LDL-c.

**Table 1 cells-11-03072-t001:** Anthropometric, clinical and biochemical characteristics of the study participants.

	Low LDL-c (n = 24)	High LDL-c (n = 36)	*p*-Value
	Median (IQR)	Median (IQR)	
Age	44.5 (36.00–53.50)	43 (35.25–50.5)	0.561
Weight (kg)	88.65 (68.25–99.20)	83.2 (73.18–92.38)	0.248
Height (m)	1.65 (1.57–1.73)	1.67 (1.56–1.74)	0.493
Waist (cm)	104.0 (90.50–114.30)	96.5 (87.25–108.0)	0.181
Hip (cm)	111.0 (103.30–125.00)	108.50 (99.50–116.00)	0.187
WHR	0.89.0 (0.81–1.02)	0.89.0 (0.79–0.99)	0.77
BMI (kg/m^2^)	31.62 (27.39–37.35)	29.86 (27.0–34.22)	0.151
PBF (%)	38.10 (32.85–44.05)	36.20 (28.80–38.70)	0.052
GLUC (mmol/l)	5.05 (4.75–5.42)	5.10 (4.80–5.74)	0.536
TGL (mmol/l)	0.95 (0.58–1.85)	0.97 (0.68–1.50)	0.689
Chol (mmol/l)	4.22 (3.70–4.68)	5.54 (5.00–6.14)	<0.0001
HDL-c (mmol/l)	1.16 (0.99–1.38)	1.25 (1.05–1.49)	0.268
LDL-c (mmol/l)	2.55 (2.30–2.70)	3.70 (3.23–4.00)	<0.0001
HbA1C (%)	5.35 (5.16–5.90)	5.70 (5.50–5.90)	0.03
Insulin (mU/L)	6.11 (5.50–10.44)	5.33 (4.89–7.32)	0.027
HOMA-IR	1.54 (1.18–2.58)	1.32 (1.11–2.17)	0.37

Abbreviations: WHR: waist to hip ratio; BMI: body mass index; PBF: percentage body fat; GLUC: glucose; TGL: Triglycrides; Chol: cholesterol; HDL: high-density lipoprotein cholesterol; LDL-c: low-density lipoprotein cholesterol; HbA1c: glycosylated hemoglobin; HOMA-IR: homeostatic model assessment of insulin resistance.

**Table 2 cells-11-03072-t002:** List of primers for qRT-PCR.

Gene	Assay ID	Gene	Assay ID	Gene	Assay ID
IL-1β	Hs01555410_m1	CCL8	Hs04187715_m1	CD141	Hs00264920_s1
IL-2	Hs00174114_m1	CCL-11	Hs00237013_m1	CD163	Hs00174705_m1
IL-5	Hs01548712_g1	CCL-15	Hs00361122_m1	CD302	Hs00994886_m1
IL-6	Hs00985639_m1	CCL18	Hs00268113_m1	TLR2	Hs01872448_s1
IL-8	Hs00174103_m1	CCL-19	Hs00171149_m1	TLR3	Hs01551078_m1
IL-10	Hs00961622_m1	CCL-20	Hs01011368_m1	TLR4	Hs00152939_m1
IL-12A	Hs01073447_m1	CXCL9	Hs00171065_m1	TLR7	Hs01933259_s1
IL-13	Hs00174379_m1	CXCL10	Hs01124251_g1	TLR8	Hs00152972_m1
IL-18	Hs01038788_m1	CXCL-11	Hs04187682_g1	TLR9	Hs00370913_s1
IL-33	Hs00369211_m1	IL-1RL1	Hs00545033_m1	TLR10	Hs01935337_s1
TNF-α	Hs01113624_g1	IL-2RA	Hs00907779_m1	IRF3	Hs01547283_m1
TGF-β	Hs00820148_g1	CCR1	Hs00928897_s1	IRF4	Hs01056533_m1
IFNB1	Hs01077958_s1	CCR2	Hs00704702_s1	IRF5	Hs00158114_m1
Dectin	Hs01902549_s1	CCR5	Hs99999149_s1	MyD88	Hs01573837_g1
SRA1	Hs00398296_g1	CD11c	Hs00174217_m1	IRAK1	Hs01018347_m1
CCL2	Hs00234140_m1	CD16	Hs04334165_m1	TRAF6	Hs00371512_g1
CCL3	Hs04194942_s1	CD68	Hs02836816_g1	NFKB	Hs00765730_m1
CCL5	Hs00982282_m1	CD86	Hs01567026_m1	GAPDH	Hs03929097_g1
CCL-7	Hs00171147_m1	IL7R	Hs00902334_m1		

**Table 3 cells-11-03072-t003:** Adipose tissue IL-23 mRNA expression correlates with multiple macrophage markers in people with H-LDL-c.

Macrophage Markers	L-LDL-c		H-LDL-c	
IL-23	r	p	r	p
ITGAX (CD11c)	0.2617	0.2651	0.5625	0.0004 *
CD16	0.4831	0.0265*	0.5944	0.0001 *
CD68	0.2429	0.2886	0.397	0.024 *
CD86	0.3158	0.175	0.3708	0.0283 *
CD127/IL7R	0.04025	0.874	0.5739	0.0004 *
CD141	−0.06028	0.7847	0.3252	0.0648
CD163	0.5325	0.0107 *	0.2899	0.0864
CD302	0.2226	0.2958	0.1278	0.4576

Abbreviation: CD: cluster of differentiation molecules found on the surface of most of the immune cells; ITGAX: Integrin alpha X. * Significant.

**Table 4 cells-11-03072-t004:** Adipose tissue gene expression of IL-23 associates with TLR2 and IRF3 in people with H-LDL-c.

	L-LDL-c		H-LDL-c	
IL-23	r	p	r	p
TLR2	0.415	0.069	0.485	0.008 *
TLR3	0.116	0.627	0.062	0.736
TLR4	0.015	0.951	0.017	0.928
TLR7	0.339	0.114	0.209	0.221
TLR9	0.150	0.483	0.024	0.891
IRF3	−0.132	0.612	0.464	0.010 *
IRF4	0.330	0.144	0.129	0.490
IRF5	0.253	0.244	0.186	0.300
MyD88	0.244	0.262	0.182	0.295
IRAK1	−0.110	0.618	0.201	0.261
TRAF6	0.390	0.059	−0.025	0.885

Abbreviations: TLR, toll like receptor; IRF, interferon regulatory factor; MyD, myeloid differentiation primary response; IRAK1, IL-1R–associated kinase; TRAF, TNF receptor associated factor. * Significant

**Table 5 cells-11-03072-t005:** Association of IL-23 with cytokines/chemokines in the people with H-LDL-c.

Cytokines/Chemokines	L-LDL-c		H-LDL-c	
IL-23	r	p	r	p
IL-1β	−0.250	0.349	0.178	0.364
IL-2	0.504	0.017 *	0.200	0.242
IL-5	−0.347	0.105	−0.220	0.235
IL-6	0.060	0.789	−0.070	0.708
IL-8	0.079	0.733	0.314	0.091
IL-10	0.209	0.363	0.097	0.581
IL-12A	−0.081	0.785	0.516	0.004 *
IL-13	0.204	0.375	0.030	0.872
IL-18	0.122	0.598	0.350	0.043 *
TNF-α	0.318	0.160	0.375	0.032 *
TGF-β	0.122	0.609	0.396	0.018 *
IFNB1	0.172	0.421	0.398	0.024 *
CCL2	0.214	0.316	0.130	0.471
CCL3	0.392	0.079	0.426	0.017 *
CCL5	0.323	0.191	0.632	0.0001 *
CCL-7	0.189	0.388	0.173	0.336
CCL-11	0.096	0.679	−0.066	0.709
CCL-15	0.163	0.457	0.370	0.026 *
CCL-19	0.023	0.920	0.078	0.656
CCL-20	0.171	0.436	0.336	0.049 *
CXCL9	0.056	0.799	0.002	0.991
CXCL10	0.162	0.460	0.205	0.245

Abbreviations: IL, interleukin; TNF, tumor necrosis factor; TGF, transforming growth factor β; CCL, C-C motif chemokine ligand; CXCL, chemokine (C-X-C motif) ligand. * Significant.

**Table 6 cells-11-03072-t006:** Association between adipose tissue IL-23 expression and metabolic parameters in people with H-LDL-c.

Metabolic Markers	L-LDL		H-LDL	
	r	p	r	p
BMI	−0.1696	0.4283	0.04376	0.7999
PBF	−0.05974	0.797	−0.1982	0.2851
Waist	−0.1702	0.4606	0.1198	0.5138
Hip	−0.1419	0.5396	0.1568	0.3916
Chol	−0.00914	0.9662	0.09003	0.6016
HDL	−0.06958	0.7466	0.3282	0.0507
LDL	0.07818	0.7165	−0.171	0.3188
TGL	−0.1591	0.4577	0.1827	0.2863
GLU	0.1319	0.5391	−0.05659	0.7431
HBA1C	0.006998	0.9741	0.1795	0.3021
WBC	−0.02482	0.9083	−0.2334	0.184
Adiponectin	−0.264	0.34	−0.437	0.0372 *

Abbreviations: BMI, body mass index; PBF, percentage body fat; GLUC, glucose; TGL, triglycerides; Chol, total cholesterol; HDL, high-density lipoprotein; LDL, low-density lipoprotein; HbA1c, glycosylated hemoglobin; WBC, white blood cells. * Significant.

## Data Availability

Not applicable.

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
