# Peer review of "Increased Adipose Tissue Expression of IL-23 Associates with Inflammatory Markers in People with High LDL Cholesterol"

_cells, 2022, doi:10.3390/cells11193072_

Round 1

Reviewer 1 Report (Previous Reviewer 1)

The manuscript resubmitted by Kochumon et al. entitled "Increased adipose tissue expression of IL-23 associates with inflammatory markers in individuals with high LDL cholesterol" was improve from the last revision. The authors aswered in a positive way to the major of the reviewers comments. In my opinion, the article reaches a suitable form to be published, however the following aspects should be improved:

- some parts of the manuscript need to be reviewed by a native English spearker;

- the definition of the table 1 needs to be improved;

Author Response

[Cells] Manuscript ID: cells-1934087

We thank the reviewer for encouraging remarks and acceptance of our manuscript following minor corrections.

Comments and Suggestions for Authors

The manuscript resubmitted by Kochumon et al. entitled "Increased adipose tissue expression of IL-23 associates with inflammatory markers in individuals with high LDL cholesterol" was improve from the last revision. The authors aswered in a positive way to the major of the reviewers comments. In my opinion, the article reaches a suitable form to be published, however the following aspects should be improved:

- some parts of the manuscript need to be reviewed by a native English spearker;

Author Response: Done

- the definition of the table 1 needs to be improved;

Author Response: Done

Reviewer 2 Report (New Reviewer)

This study aims to find the role of adipose IL-23 expression as a potential inflammatory marker in people with high LDL cholesterol. Overall, the study was well-performed and presented. Even so, some revisions are needed.

A major one is isotype controls were not shown or listed as supplementary data.

Some minors:

Typo errors should be taken care of, such as periods in lines 181 and 204 and Figure 23.

20X > 20× (line 189) and P value or p value should be consistent.

Abbreviations such as TNF-α and HOMA-IR.

Table 1, reforms the lines in column 2.

Figure 1A, LDL < 2.9 and LDL > 2.9 changed to L-LDL-c and H-LDL-c to be consistent with the following figures.

Author Response

Response to the Reviewer 2 comments

[Cells] Manuscript ID: cells-1934087

We thank the reviewer for encouraging remarks and acceptance of our manuscript following minor corrections.

Comments and Suggestions for Authors

This study aims to find the role of adipose IL-23 expression as a potential inflammatory marker in people with high LDL cholesterol. Overall, the study was well-performed and presented. Even so, some revisions are needed.

A major one is isotype controls were not shown or listed as supplementary data.

Author Response:  Isotype controls are now included in the supplementary figure S1.

Some minors:

Typo errors should be taken care of, such as periods in lines 181 and 204 and Figure 23.

20X > 20× (line 189) and P value or p value should be consistent.

Author response: Done

Abbreviations such as TNF-α and HOMA-IR.

Author response: Done

Table 1, reforms the lines in column 2.

Author response: Done

Figure 1A, LDL < 2.9 and LDL > 2.9 changed to L-LDL-c and H-LDL-c to be consistent with the following figures.

Author response: Done

This manuscript is a resubmission of an earlier submission. The following is a list of the peer review reports and author responses from that submission.

Round 1

Reviewer 1 Report

The manuscript submitted by Kochumon et al to Cells-MDPI entitled "Increased adipose tissue expression of IL-23 associates with inflammatory markers in individuals with high LDL cholesterol" is very well-written and is relevant to the field of Clinical Nutrition and Surgery. Although, the subject of the study (IL-23) is not new, the results and the discussion provided is solid and supported by previous works.

In the line 417, the authors should remove the typo ")." 

Of course that this reviewer, would like to appreciate to see an additional evaluation using the intra-abdominal fat tissue. However, the present study is so far relevant and complete.

Reviewer 2 Report

1. Table 1. The values are represented as mean + SD, should be presented as Median (IQR)

2. Legends to Table1 should be added

3. Methods section, subsection 2.3 to be replaced as Measurement of Biochemical parameters

4. The authors have to present protein data with the RNA gene data?

5. Volcano plots represents better gene correlation results?

Reviewer 3 Report

In the manuscript ‘Increased adipose tissue expression of IL-23 associates with inflammatory markers in individuals with high LDL cholesterol’ by Shihab Kochumon et al., the authors evaluate relationships between plasma LDL cholesterol and IL-23 gene expression in the adipose tissue of non-diabetic participants. Their main finding is that, indeed, such relationship exists. The study deals with a potentially important issue of clarifying the links between dyslipidemia, inflammation and atherosclerosis.

It is generally well designed and performed. The aim is clear, and the methodological tools appropriate. The manuscript is well written and easy to follow.

My major concern relates to the credibility and significance of the obtained results.

As for the credibility, the authors state that IL-23 gene expression associates positively with plasma concentrations of LDL-cholesterol. Indeed, the r-value for this association equals 0.471. However, as seen in Figure 1, much of this relationship depends on the single outlier with no IL-23 gene expression whatsoever. More importantly, in Table 6, IL-23 shows absolutely no association with LDL cholesterol in the two subgroups evaluated separately. This casts in doubt the main finding of the study.

As for the significance, this is a cross-sectional study. In order to drive conclusions presented in the paper, it should be supported by other analyses, interventional that would allow to speculate on causal relationships.

As for the other findings, IL-23 is an inflammatory cytokine. No wonder, it associates with other cytokines. Their expressions and levels increase during an inflammatory state in parallel, so the positive associations between them are obvious.

Minor points;

Information on lipid-lowering drugs is essential. Statins obviously decrease LDL-c, while their impact on IL-23 gene expression is unknown, probably mild. At best, participants on statins ought to be excluded from the analysis. At least, the information on lipid lowering therapy in the studied group should be provided.  

In the abstract, comparative data should be given, not only p-values.